# Colored Shade Nets Can Relieve Abnormal Fruit Softening and Premature Leaf Senescence of “Jumeigui” Grapes during Ripening under Greenhouse Conditions

**DOI:** 10.3390/plants11091227

**Published:** 2022-05-01

**Authors:** Qian Zha, Xiangjing Yin, Xiaojun Xi, Aili Jiang

**Affiliations:** 1Research Institute of Forestry and Pomology, Shanghai Academy of Agricultural Sciences, Shanghai 201403, China; zhaqian@saas.sh.cn (Q.Z.); yinxiangjing@saas.sh.cn (X.Y.); 2Shanghai Key Labs of the Protected Horticultural Technology, Shanghai Academy of Agricultural Sciences, Shanghai 201403, China

**Keywords:** abnormal softening, colored shade net, diurnal temperature, grape, premature leaf senescence

## Abstract

High temperature causes premature grape leaf senescence, abnormal berry softening, and shortening of the fruiting period. Furthermore, the fruit quality and yield are severely affected. Here, the “Jumeigui” grape quality and leaf senescence were evaluated under shading; green, blue, black, and gray nets were used for shading, and their spectra were measured. At the same density, the shade-net color significantly affected cooling and shading efficiencies, with gray nets showing the best light transmission and cooling effect. Shading significantly alleviated abnormal heat-induced grape softness. The total soluble solids (TSS) content and grape coloration were affected under gray, blue, and green shade nets. Nonetheless, TSS exceeded 18 °Brix under gray, blue, and green nets, as required of first-class high-quality fruit. The peel color was not significantly affected under gray or blue shade nets, whereas unshaded grapes showed clear heat-stress damage, especially on the edges of unshaded bottom leaves, in which the net photosynthesis rate was significantly lower than that under shading, indicating that high light intensity and heat caused premature leaf senescence. Colored shade nets reduced greenhouse temperature and light intensity, thereby alleviating the premature senescence of grape plants. Grape quality under black shade nets was poor, whereas superior quality was achieved using gray or blue shade nets.

## 1. Introduction

Recently, the area under grape cultivation in China has significantly expanded with the vigorous promotion of cultivation techniques, such that Southern China has become the main production area for table grapes. The optimum temperature range for grape development is 25–35 °C [1]. However, the grape ripening phase in greenhouse facilities in Southern China usually takes place during an extremely hot period (~45 °C) in the open field [2]. The effect of the sun on the composition of grapes can be significant and complex [3]: solar radiation and heat can affect the rate of metabolic reactions indirectly via dehydration, or directly, via heat stress.

*Vitis vinifera* variety L. “Jumeigui” was selected and bred in the Grape Variety Garden of Dalian Academy of Agricultural Sciences in 1993 with ‘Shenyang big Muscat’ as the female parent and “Kyoho” as the male parent. In China, “Jumeigui” is widely cultivated for table grape production [4], as it shows high fruit setting rate, a strong fragrance, and moderate sweetness and sourness; however, it is also prone to experiencing extremely hot periods at maturity, resulting in abnormal fruit softening [5] and premature senescence of the leaves, which in turn shortens the fruit-bearing period, ultimately severely affecting fruit quality and plant productivity. Many cooling methods and technologies have been proposed to reduce the effect of sunlight during crop production. Thus, for example, watering can reduce canopy temperature by 7–10 °C, thus alleviating high temperature damage [6]. However, watering increases air humidity, easily leading to disease incidence. Alternatively, the use of shade nets is the most widely used approach, due to its low cost and simple operation [7]. Nonetheless, if the shade net type is not carefully selected and does not allow plants to maintain effective photosynthesis during the cooling process, shading will also affect plant growth [8] and reduce crop productivity. Recently, colored shade nets were introduced and have become widely adopted. These nets can filter the sunlight spectrum and disperse direct light through a special component that promotes crop growth [9]. Therefore, temperature, humidity, and light in the growth environment show interactive effects [10] whereby they should be considered to comprehensively assess fruit growth and development.

Warren et al. [11] reported that the growth and yield of horticultural crops in growth facilities can be strongly dependent on light, and it is generally understood that shading affects grape quality [12]. The use of shading nets will reduce light intensity, such that the incorrect use of shading nets can have a negative impact on growth and production [8]. Hence, here we aimed to use colored shade nets to identify sunshade materials with a higher cooling effect, concomitant with high light-transmittance properties to achieve high-quality grape production. Our findings are particularly important given the context of current climate change, which is causing global temperatures to rise.

## 2. Results

### 2.1. Temperature and Light Transmittance

Unshaded plants were frequently exposed to 40 °C or higher ambient temperatures from July to September in the greenhouses of the grape growing region of Southern China. In contrast, the temperature in the greenhouse was significantly reduced when colored shade nets were used (Figure 1A). Table 1 summarizes the ambient temperature differences between unshaded and shaded treatment areas. The black shade net produced the largest decrease in minimum and maximum temperatures (0.8 and 7.6 °C, respectively), followed by gray, blue, and green nets, with minimum and maximum temperature declines of 1.2 and 7 °C, 0.7 and 6.1 °C, and 0.4 and 6.1 °C, respectively. These data indicate that the cooling effects of the black and gray shade nets were greater than those of either blue or green nets. Concomitantly, there was no significant difference in relative humidity among treatments (Appendix A). The determination of absolute spectral value of the colored shade nets in the wavelength range of 300–800 nm is shown in Figure 1B; the strongest absorption regions of chlorophyll were 430–450 nm and 640–660 nm, where the absolute spectral value of colored shade nets was significantly reduced. Light transmittance (300–800 nm) was higher under the blue (57.3%) and gray (46.2%) shade nets than under the green (40%) or black (17.1%) shade nets (Table 2). Leaves were prone to withering and premature aging in the absence of shading, which affected grape appearance (Appendix A). Peel color was best for unshaded berries, followed by that observed in berries growing under blue and gray shading nets, with the poorest berry color observed for berries growing under the black shade net (Figure 2A,E).

### 2.2. Berry Internal and External Properties

We used conventional methods to determine fructose, glucose, and anthocyanin contents (Figure 2). The results showed that they were all reduced to varying degrees at the two study sites, indicating that the shading treatment affected grape appearance and sugar content. However, these data only included three non-biological replicates; each replicate was a mixture of 10 samples, and the average value was determined. Furthermore, 30 separate values from 30 berries were selected to determine TSS, CIRC, and other parameters. The data were objectively represented as a frequency distribution.

Based on the method used to evaluate disease and heat damage, we designed a method to subjectively evaluate fruit softening, expressed as an index (Table 3), such that the higher the softening index, the softer the fruit. Further, ’Jumeigui’ berries became abnormally soft in the absence of shading, with a fruit softening index ranging from 72% to 73.33%. In contrast, colored shade nets significantly alleviated abnormal fruit softening. Thus, the softening index decreased to 23.33%–31.33% and 35.33%–59.33% at PQ and ZH, respectively. Subsequently, a texture analyzer was used to evaluate fruit firmness.

We generated normal distribution diagrams of the values recorded for grape TSS contents (Figure 3A,D). Shifting of the curve further to the right indicates a higher TSS value, while a higher curve apex indicates greater consistency in TSS. The minimum TSS required for high-quality fruit is 18 °Brix. Although the TSS of the shaded grapes decreased, it did reach 18 °Brix under the gray, blue, and green shade nets, indicating that these shade treatments produced high-quality fruits. Moreover, the normal distribution curve for TSS peaked for the gray shade treatment, indicating that TSS was more consistent under this treatment.

Similarly, we generated normal distribution diagrams of the peel color index (Figure 3B,E). In this case, a normal distribution diagram shifted to the right indicates a more intense red peel color, while a higher curve apex indicates higher color consistency. Hence, the peel color was affected by shading, and the black shade net had the greatest influence on the peel color index. A darker red peel gives “Jumeigui” a better appearance. Conversely, the gray shade net had the smallest effect on peel color, particularly at PQ.

Fruit firmness showed clear differences across treatments (Figure 3C,F). Thus, at PQ, fruit firmness was highest under the blue shade net, followed by gray, black, green, and the unshaded treatment, and was more consistent under the blue shade net than under any other shade treatment. Similarly, at ZH, fruit firmness was lower under the unshaded treatment than under any shading treatment. 

### 2.3. Leaf Photosynthetic Performance

We selected five bottom and five top leaves (Appendix A) to determine leaf *Ci*, *A*, *gs*, and *E* values under the various shading treatments described above. At PQ, top leaf *Ci* was significantly lower under the gray and blue shade nets than under any other treatment (Figure 4A). At the same time, *A* was significantly lower in bottom leaves of plants kept under unshaded conditions than in those of plants growing under the shading treatments (left side of Figure 4B); furthermore, neither *gs* nor *E* differed significantly among the five treatments (left side of Figure 4C,D). Further, while top leaf *A* did not differ significantly among treatments (Figure 4B), *Ci*, *gs*, and *E*, were all significantly lower in top leaves under gray and blue shade nets than under the other treatments (Figure 4A,C,D). In contrast, at ZH, bottom leaf *A* was significantly lower, and the *Ci* was significantly higher, under the unshaded than under the shading treatments (left side of Figure 4E,F). Meanwhile, *gs* and *E* values in the bottom leaves, and those for *Ci*, *A*, *gs*, and *E* in top leaves showed no significant differences across the five treatments (Figure 4E–H), indicating that the shading effects were greater on top leaves.

At PQ, the *F*_v_/*F*_m_ and *PI*_ABS_ values for all leaves measured were significantly lower in the unshaded plants than in those growing under gray or black shade nets (Figure 5A,B). In contrast, *W*_k_ and *M*_o_ were significantly higher in unshaded than under the shaded plants (Figure 5C,D). Meanwhile, leaf *Ψ*_Eo_ and *Φ*_Eo_ values were significantly lower under the unshaded treatment than under the shading treatments (Figure 5E,F), and *δ*_Ro_ in all leaves measured was highest under the unshaded treatment, and significantly higher than the corresponding values in plants under black shade nets (Figure 5G).

As for the study site at ZH, the *F*_v_/*F*_m_ and *PI*_ABS_ values in all leaves measured were significantly higher in the shade-treated than in the unshaded control plants, and peaked in plants under a black shade net (Figure 5H,I). Conversely, *W*_k_ and *M*_o_ were significantly higher in the unshaded plants than in any shade-treated plants but especially in those under black or gray shade nets (Figure 5J,K). Furthermore, *Ψ*_Eo_ and *Φ*_Eo_ were significantly lower in the unshaded plants than in those under black, gray, or green shade nets (Figure 5L,M). Finally, *δ*_Ro_ values in the bottom leaves were significantly lower in plants growing under the black shade net than in those growing under any the other treatment, while *δ*_Ro_ values for top leaves did not significantly differ among the five treatments (Figure 5N).

## 3. Discussion

According to climate model predictions, the grape growing season will become warmer, while some cool climate regions are expected to become warm or hot regions by 2080 [13]. Climate warming has also led to rapid grape ripening and leaf aging [14]; however, shading can attenuate this problem, as colored shade nets show differential spectral transmittance and reflection properties [8]. Indeed, these nets are being increasingly and extensively adopted, and have shown to improve the microclimate [15], thereby providing an effective strategy for greenhouse crop production.

### 3.1. Shading Can Relieve Leaf Damage Due to Enhanced Photosynthesis

Chlorophyll is critical in the photosynthetic process. Different light wavelengths interact with the corresponding photoreceptors to regulate the synthesis of different pigments [16]. The absorption wavelength of visible light is mainly concentrated in the 430–450 and the 640–660 nm regions of the spectrum [17]. Therefore, we focused on analyzing the light transmittance of the strongest chlorophyll absorption area. Gray and blue shade nets have higher light transmittance in those regions, thereby allowing for the growth of healthier leaves (Appendix A).

Wu et al. [18] found that shading can prevent summer heat-stress damage in grape plants. In our study, we examined fruit vines and used a phased shading treatment to achieve ideal cooling and light transmission. We found that *A* values of the five bottom leaves were higher under shaded than under unshaded plants, indicating that prolonged heat and high light-intensity stress can result in premature leaf aging. Further, the trends in chlorophyll fluorescence parameters and photosynthetic indicators differed, reflecting an immediate plant response to stress. Thus, all leaf *W*_k_ and *PI*_ABS_ values under shading were higher than those under the unshaded treatment. Furthermore, as a key indicator of heat stress, *W*_k_ is higher at high temperature, because heat damages the oxygen-evolving complex [19]. In contrast, *PI*_ABS_ decreases after heat treatment, reflecting the inhibition of PSII activity [2]. In this study, we found that unshaded leaves showed the highest *W*_k_ values and low *PI*_ABS_ values, indicating the occurrence of heat stress in the absence of shading, whereas shading had a positive effect on leaf photosynthesis.

### 3.2. Shading Can Reduce Fruit Damage Due to Heat-Induced Softening, and Ensure Fruit Quality

Light and temperature can affect sugar accumulation in wine and table grapes [20]. In our study, the soluble-solid content decreased slightly during the shading treatment, but, on average, remained > 18 °Brix, which satisfies the requirements for high-quality fruit, indicating that shading reduces berry sugar content without affecting its quality.

Anthocyanin content is influenced both by environmental factors and viticulture management practices. Several studies in grapes have reported the effects of light exposure or shading on anthocyanin accumulation [21,22]. Temperature is another important factor that affects anthocyanin biosynthesis in plants; specifically, the expression of anthocyanin biosynthesis-related genes is reportedly inhibited by heat stress [23,24]. This indicates that temperature and light are major factors in determining the grape peel color. Hence, applying the appropriate temperature and light is crucial to achieve the best berry skin color. Therefore, in this study, we aimed to determine the conditions that reduce ambient temperature without causing excessive shading, thereby providing the most suitable environment for grape growth and development. We found that gray and blue shade nets were ideal for optimum peel coloring.

The effect of shade nets on fruit development and quality depends mainly on the cultivar, the environment, and the light utilization efficiency at the canopy level [25]. Gray shades scatter light to a greater extent, resulting in a more even light distribution on plants [26], thereby improving light use efficiency, photosynthetic efficiency, and dry matter accumulation [27]. For instance, tomato yields are grown under gray shade nets [15], which guard crops from exposure to direct sunlight and prevent over-ripening [28]. Therefore, the use of shade nets delays both harvesting and ripening and positively affects grape quality. In this study, the berry sugar content was lower and the peel color was less intense in shaded than in unshaded plants, thus reflecting delayed maturity, consistent with the previous reports [29,30]. 

## 4. Materials and Methods

### 4.1. Plant Material and Treatments

The experiments reported herein were performed from July to September 2020, at the Zhuanghang (ZH) experimental vineyard of the Shanghai Academy of Agricultural Sciences, Shanghai, China (30°51′ N, 121°13′ E) and at the Shanghai Pingqi (PQ) Grape Planting Professional Cooperative (30°11′ N, 121°25′ E), using the table grape cultivar “Jumeigui”. The study site has a subtropical monsoon climate. Due to the high rainfall in this area, grapes are planted under plastic film cover to reduce the incidence of plant diseases and insect pests. Controlled environment agriculture (shè shī nóng yè in Chinese) is an agricultural production strategy that uses clustered plastic-roofed greenhouses and irrigation for crop production [31]. Self-rooted grapevines were planted in the spring of 2008 in a north–south orientation at a 4 m × 2.8 m spacing. The vines were grown in a rain shelter with a Y-shaped training system at ZH (8 m × 45 m) and a flat scaffolding system at PQ (6 m × 41 m). The light-exposed leaf area (m^2^/m^2^ of ground area) was estimated using Carbonneau’s method [32] only once per treatment. Therefore, the leaf-to-fruit ratio was 15 leaves to ensure one fruit bunch, and the weight of the bunch berries was 0.5~0.6 kg. Through tree management, the yield of grapevines in this experiment was controlled at 1000 kg/667 m^2^. A lime–sulfur mixture was applied for disease control before germination. Pesticide control of *Botrytis cinerea* was applied during flowering. Comprehensive disease prevention was applied before berry bagging.

Blue, green, and black shade nets, and aluminum foil gray shade nets were purchased from an agricultural market (Jiehang Agricultural Materials Factory, Yangzhou, Jiangsu, China). The colored shade nets (6 m × 41 m at PQ, and 8 m × 45 m at ZH) were used to cover single-shed facilities. The experiment was laid in a randomized complete block design with three replicates. For control treatment, no shade net was used. The treatment period was from 24 July 2020 (50% color change) to 3 September 2020. The fruit was harvested on 18 August 2020, and leaf-related indicators were measured on 3 September 2020. Grapes were judged to be mature when the total soluble solid content (TSS) was higher than 16 °Brix. Grape harvest was performed when the grapes were mature, and measurements of the leaf photosynthetic performance were conducted after harvest. An RC-4HC temperature/humidity Data Logger Recorder (Jiangsu Jingchuang Electric Co., Ltd., Nanjing, China) placed near the berries was used to record the temperature. All experimental and analytical methods were performed in accordance with the relevant guidelines and regulations. 

### 4.2. Spectral Analysis

A PMS-2000 UV-VIS-near IR spectrophotocolorimeter (Everfine Co., Ltd., Hangzhou, China) was used to analyze the physical properties of transmitted light. Spectral data analysis of transmitted light was performed under different color shading nets, and the wavelength range used for analysis was 300–800 nm.

### 4.3. Total Soluble Solids (TSS)

For each treatment, 30 berries from 10 top, middle, and bottom bunches were prepared, and TSS contents were determined after pressing the juice separately. TSS was measured in °Brix using a PAL-1 digital refractometer (Atago, Tokyo, Japan).

### 4.4. Fast Chlorophyll a Fluorescence Kinetic Parameters

A Pocket-PEA fluorimeter (Plant Efficiency Analyzer, Hansatech Instruments Ltd., King’s Lynn Norfolk, UK) was used to determine fast leaf chlorophyll *a* fluorescence-kinetics [33]. The definitions of the fluorescence parameters are summarized in Appendix A.

### 4.5. Leaf Gas-Exchange Parameters

Photosynthesis measurements were performed on five leaves randomly selected near the base (hereafter referred to as bottom leaves, Appendix A) and five more near the top (hereafter referred to as top leaves, Appendix A). Measured gas-exchange parameters included net photosynthesis rate (*A*), stomatal conductance (*gs*), transpiration rate (*E*), and leaf intercellular CO_2_ concentration (*Ci*). These parameters were all measured using a portable photosynthetic CIRAS-3 system (PP Systems, Amesbury, MA, USA) [34].

### 4.6. Sugar Analysis

Ten berries were selected from the three top, middle, and bottom bunches, and pooled. Berry pulps were ground under liquid nitrogen prior to further use. Each treatment was replicated three times. The extraction of soluble sugars was performed and determined via high-performance liquid chromatography (HPLC) as described by Zha et al. [14]. Soluble sugars were extracted from 3 g of frozen berry powder homogenized in 6 mL ethanol/water (4:1 *v*/*v*) at 35 °C for 20 min. The homogenate was centrifuged at 6500× *g* for 15 min, and the residues were re-extracted using the same procedure. The supernatants from the two extractions were mixed and made up to 15 mL with distilled water. Thereafter, 1 mL of the extract was evaporated under vacuum at 35 °C, redissolved in 1 mL MilliQ water (MilliporeSigma, Burlington, MA, USA), and passed through a 0.45 μm Millipore filter. Soluble sugar contents were determined using the Waters E2695 system (Waters, Milford, MA, USA).

### 4.7. Anthocyanin Content

Ten berries were selected from the three top, middle, and bottom bunches and pooled. Berry skins were ground under liquid nitrogen prior to further use. Each treatment was replicated three times. Anthocyanin content was determined using the pH-differential method [35,36].

### 4.8. Classification of Abnormal Softening Symptoms

We examined abnormal grape softening in the field and developed a grading standard based on which we investigated 30 bunches per treatment and estimated the softening percentage using specific equations. Softening was graded according to the following scale: level 0, no abnormal softening symptoms (N1); level 1, abnormal softening up to 10% (N2); level 2, abnormal softening up to 30% (N3); level 3, abnormal softening up to 50% (N4); level 4, abnormal softening up to 70% (N5); and level 5, abnormal softening up to 100% (N6).
Softening index = (0 × N1 + 1 × N2 + 2 × N3 + 3 × N4 + 4 × N5 + 5 × N6)/[5 × (N1 + N2 + N3 + N4 + N5 + N6)](1)

Numbers 0 to 5 indicate plant replicate number.

### 4.9. Grape Texture

Thirty berries from ten of the top, middle, and bottom bunches were prepared per treatment, and the berry texture was analyzed using a TA.XT.Plus type physical property tester (Stable Micro System, Godalming, UK). Fruit firmness was expressed as the force (N) required to deform the berries, according to Lijavetzky et al. [37]. 

### 4.10. Berry Skin Color

Thirty berries from ten top, middle, and bottom bunches were prepared, and the berry color at the equatorial portion of each berry was recorded using a hand-held C410 Chroma Meter (Konica Minolta, Chiyoda-ku, Tokyo, Japan). Color index of red grapes (CIRG), which is based on CIELab data, was calculated using the formula CIRG = (180 − *h*)/(*L** + *C**) [15]. The index used for the evaluation of the appearance and color of the berries was: CIRG < 2 represents yellow-green, 2 < CIRG < 4 pink, 4 < CIRG < 5 red, 5 < CIRG < 6 deep red, and CIRG > 6 blue-black [38]. 

### 4.11. Statistical Analysis

Total soluble solids and CIRG data were tested to confirm the normality of distribution. These data were compared through two dimensions, average and distribution. The measurements were arranged in ascending order and divided into several groups according to group interval (*h*). The number of measurements in each group was called the frequency of the group. Then, frequency distributions were generated using the function “NORMDIST” in Microsoft Excel 2010 (Redmond, WA, USA).
*h* = (*x_n_* − *x*_1_)/(*k* − 1)(2)
where, *x_n_* is the upper limit of coordinate axis; *x*_1_ is the lower limit of coordinate axis; and *k* is the number of groups.

The differences among treatments were assessed using one-way ANOVA followed by Tukey’s test using SPSS v22.0 (IBM Corp., Armonk, NY, USA). 

## 5. Conclusions

The use of colored shade nets during the grape ripening period effectively reduced temperature and light intensity in the greenhouse. Leaf senescence rates at the base of the grape branches were significantly reduced. After shading, the fruit remained firm, and the fruiting period was extended. The shading treatment had a positive effect on the growth of grape plants. Among the nets tested, compared with black and green, gray and blue shade nets resulted in greater sugar accumulation in berries and greater consistency of fruit coloring. Therefore, gray and blue shade nets have greater application potential in viticulture. 

## Figures and Tables

**Figure 1 plants-11-01227-f001:**
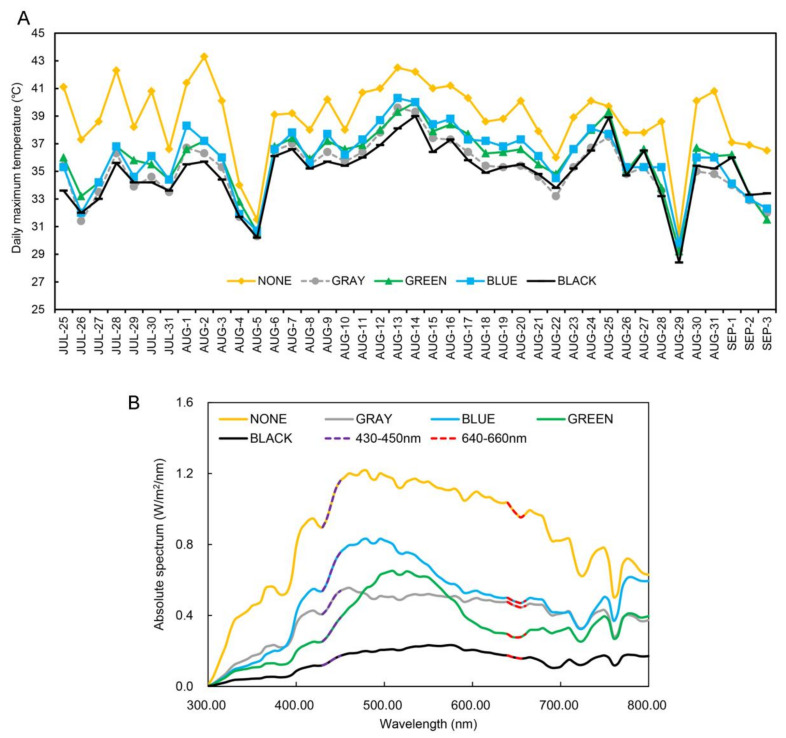
Processing temperature and spectrum data in the facility greenhouse under the different shading treatments. (**A**) The daily maximum temperature trend in the facility greenhouse under the colored shading materials. (**B**) The projected light spectrum composition of the colored shading materials. NONE: no shading; GRAY, BLUE, GREEN, and BLACK: shade net colors.

**Figure 2 plants-11-01227-f002:**
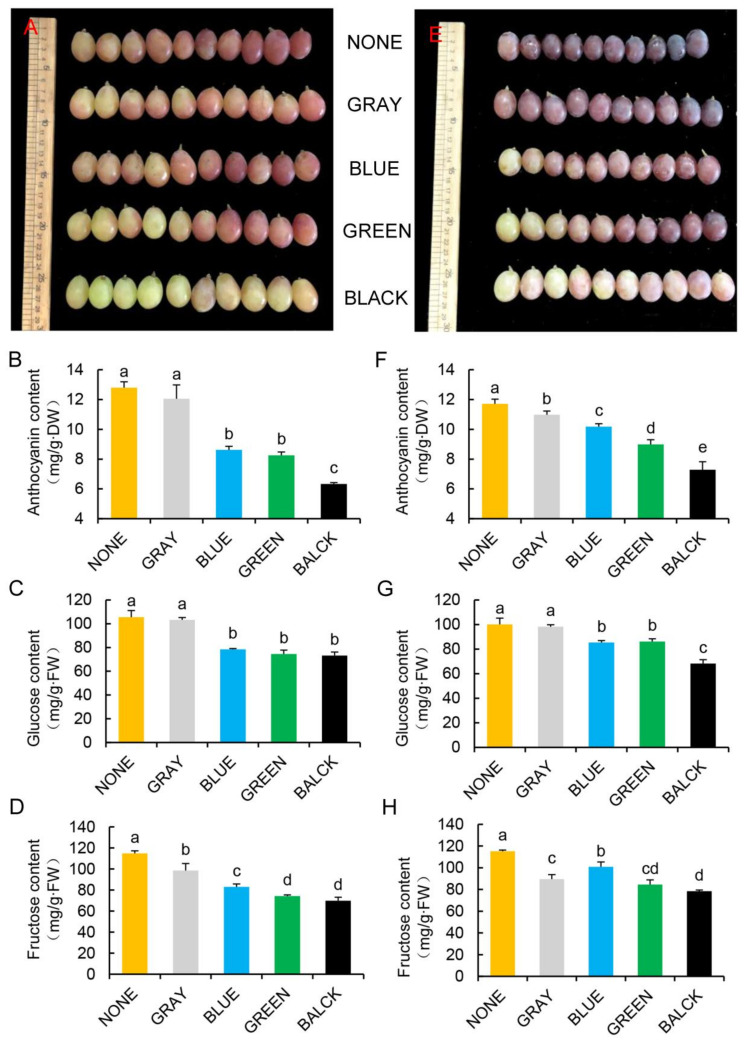
The grapes and their composition across the different treatments. (**A**,**E**): grape images; Anthocyanin, glucose, and fructose levels at PQ (**B**–**D**) and ZH (**F**–**H**). NONE: no shading; GRAY, BLUE, GREEN, and BLACK: shade net colors. Different lowercase letters indicate significant differences (*p* < 0.05).

**Figure 3 plants-11-01227-f003:**
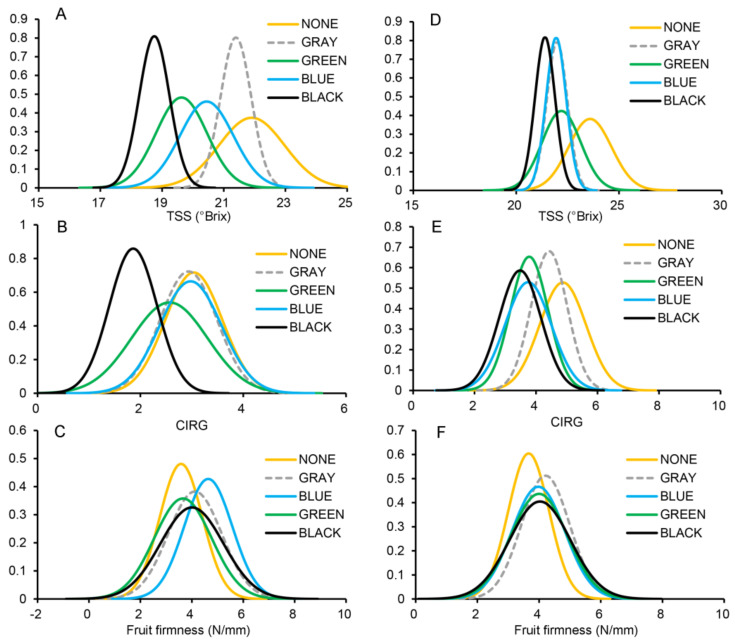
Differences in TSS, CIRG, and fruit firmness across the different treatments. TSS, CIRG, and fruit firmness, at PQ (**A**–**C**) and ZH (**D**–**F**). NONE, no shading; GRAY, BLUE, GREEN, and BLACK, shade net colors.

**Figure 4 plants-11-01227-f004:**
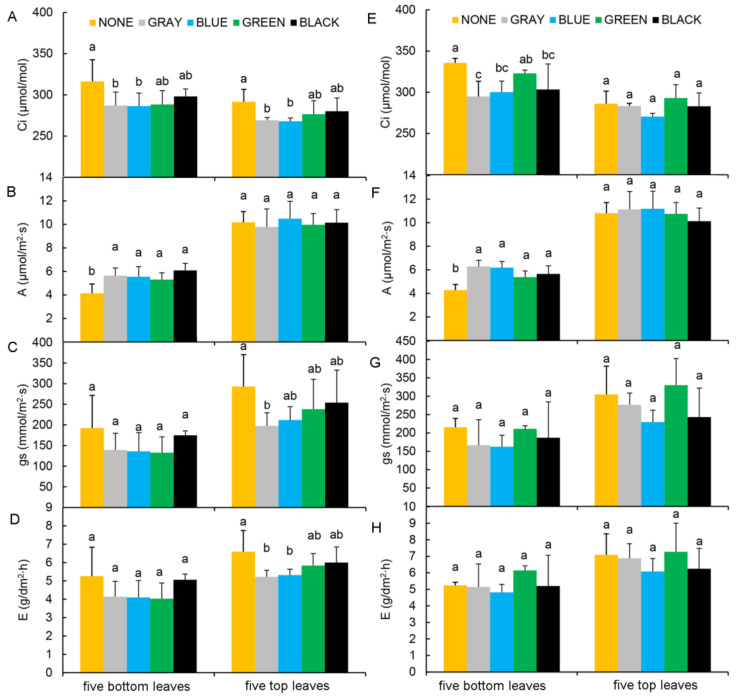
Differences in the photosynthesis-related parameters of the grapevine leaves across the different treatments. *Ci*, *A*, *gs*, and *E* across the different treatments at PQ (**A**–**D**) and ZH (**E**–**H**). NONE, no shading; GRAY, BLUE, GREEN, and BLACK, shade net colors. Different lowercase letters indicate significant differences (*p* < 0.05).

**Figure 5 plants-11-01227-f005:**
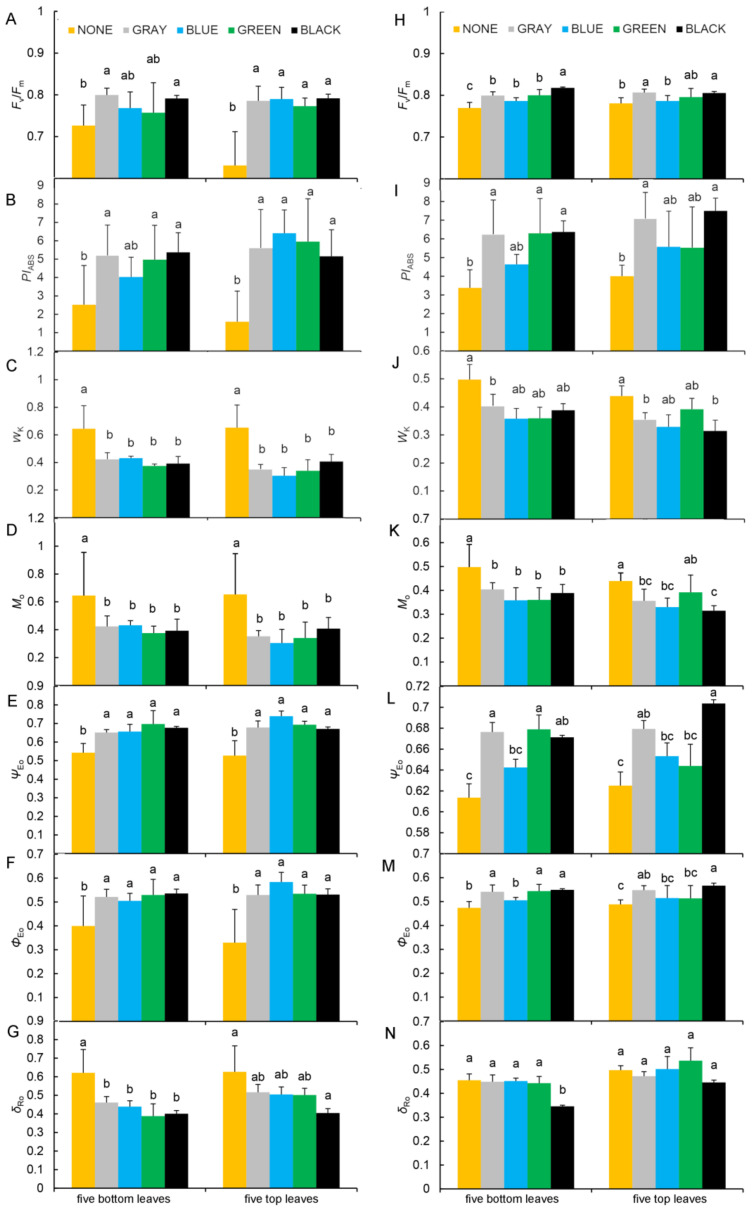
Differences in PSII-related parameters of grapevine leaves under the different treatments. *F*_v_/*F*_m_, *PI*_ABS_, *W*_k_, *M*_o_, *Ψ*_Eo_, *Φ*_Eo_, and *δ*_Ro_ at PQ (**A**–**G**) and ZH (**H**–**N**). NONE, no shading; GRAY, BLUE, GREEN, and BLACK, shade net colors. Different lowercase letters indicate significant differences (*p* < 0.05).

**Table 1 plants-11-01227-t001:** Temperature difference between the unshaded and the shaded treatments (gray, green, blue, or black net).

Temperature (°C)	GRAY	GREEN	BLUE	BLACK
∆min	1.2	0.4	0.7	0.8
∆max	7	6.1	6.1	7.6

**Table 2 plants-11-01227-t002:** Light transmittance of the colored shade net (gray, green, blue, or black) at different wavelengths. This was calculated by dividing the sum of the values of the shading treatment transmission spectra in the 430–450 nm range by the sum of the values of the solar spectrum in the same range.

Wavelength Range (nm)	GRAY	GREEN	BLUE	BLACK
300–800	46.2	40.00	57.3	17.1
430–450	46.1	30.82	63.0	14.1
640–660	46.4	28.84	48.9	16.6

**Table 3 plants-11-01227-t003:** Softening index of the “Jumeigui” grape cultivar.

Treatment	Softening Index (%)	
	PQ	ZH
NONE	73.33	72
GRAY	26.67	43.33
BLUE	23.33	42.67
GREEN	31.33	59.33
BLACK	26	35.33

NONE: no shading; GRAY, BLUE, GREEN, and BLACK: shade net colors. PQ and ZH: experimental vineyard sites at the Shanghai Pingqi Grape Planting Professional Cooperative (30°11′ N, 121°25′ E), and at Zhuanghang, the site at the Shanghai Academy of Agricultural Sciences, Shanghai, China (30°51′ N, 121°13′ E), respectively.

## Data Availability

Data is contained within the article.

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
