# Peer review of "Colored Shade Nets Can Relieve Abnormal Fruit Softening and Premature Leaf Senescence of “Jumeigui” Grapes during Ripening under Greenhouse Conditions"

_plants, 2022, doi:10.3390/plants11091227_

Round 1

Reviewer 1 Report

Dear authors and editor,

the manuscript “Colored Shade Nets can Relieve Abnormal Fruit Softening and
Premature Leaf Senescence of “Jumeigui” Grapes during Ripening Under Greenhouse Conditions
” aims at investigating the effects of different-colours shading nets on the alleviation of heat and light stress and their consequences on vines senescence. The authors performed a reflectance analysis to understand the effects of different treatments on chlorophyll and senescence. However, the focus is clear only in the Discussion. Therefore, the sections must be re-organised to emphasise the objectives.

The Introduction is poor with references. The effects of heat and light stress on the physiological mechanisms and the berry quality must be addressed. There are some recent references you can refer to (https://doi.org/10.3390/agronomy11101940, https://doi.org/10.1111/plb.12410). The gaps in previous research are completely missing. Some references to the parameters that you have analysed and their importance must be included (especially the reflectance data). After clarifying these points, you will be able to state clear objectives.

In M&M you don’t provide information on the size of the experimental area and each treatment. Moreover, some missing in the statistical analysis must be filled (see specific comments). I don’t understand why you have monitored the physiological parameters on old leaves (see specific comments). Please, provide some additional explanation.

The results are confused and confusing. It looks like there is a decoupling within the results of different tests. Capturing the opportunity of using one net or another is quite difficult. You should address better your findings for practical applications.

The Discussion is more a summary of the findings than a critical analysis. As the results are pretty confusing, you should drive here what can be learned from your findings.

Here are some specific comments:

Line 124-125: which specific equations? You should report them and their scientific background

Line 148: why only TSS and CIRG?

Lines 187-195: not appropriate for the results section. Describe your results here and move these lines to M&M

Line 228: why old leaves? Mature leaves should be selected. It is known that leaves older than 5-6 weeks weaken their photosynthetic rate

Author Response

Reviewer 1

The manuscript “Colored Shade Nets can Relieve Abnormal Fruit Softening and
Premature Leaf Senescence of “Jumeigui” Grapes during Ripening Under Greenhouse Conditions
” aims at investigating the effects of different-colours shading nets on the alleviation of heat and light stress and their consequences on vines senescence. The authors performed a reflectance analysis to understand the effects of different treatments on chlorophyll and senescence. However, the focus is clear only in the Discussion. Therefore, the sections must be re-organised to emphasise the objectives.

The Introduction is poor with references. The effects of heat and light stress on the physiological mechanisms and the berry quality must be addressed. There are some recent references you can refer to (https://doi.org/10.3390/agronomy11101940, https://doi.org/10.1111/plb.12410). The gaps in previous research are completely missing. Some references to the parameters that you have analysed and their importance must be included (especially the reflectance data). After clarifying these points, you will be able to state clear objectives.

Response: Thank you for insightful comments. Relevant references have been added in the introduction section.

In M&M you don’t provide information on the size of the experimental area and each treatment. Moreover, some missing in the statistical analysis must be filled (see specific comments). I don’t understand why you have monitored the physiological parameters on old leaves (see specific comments). Please, provide some additional explanation.

Response: Thank you for these comments. The information on the size of the experimental area and each treatment has been provided in section 2.1. The old leaves described in the manuscript are actually the five lower leaves near the base on a fruiting branch (Figure S1), which has been more accurately described in the revised manuscript.

The results are confused and confusing. It looks like there is a decoupling within the results of different tests. Capturing the opportunity of using one net or another is quite difficult. You should address better your findings for practical applications. The Discussion is more a summary of the findings than a critical analysis. As the results are pretty confusing, you should drive here what can be learned from your findings.

Response: We apologize that the description of the results was confusing. Gray and blue shade nets had a better effect on grape fruit quality improvement than black and green shade nets, which may be related to their light transmittance and spectral data. This is the focus of our follow-up work, and more experiments need to be designed.

 Here are some specific comments:

 Line 124-125: which specific equations? You should report them and their scientific background

Response: The related information has been added in section 2.11 of revised manuscript.

Line 148: why only TSS and CIRG?

Response: TSS and CIRG of each treatment have 30 separate values in our experiment, which can be used for frequency distribution. The frequency distribution can not only display the average value, but also the distribution of the data. Therefore, this method could be better applied to evaluate fruit quality in cultivation experiments.

Lines 187-195: not appropriate for the results section. Describe your results here and move these lines to M&M

Response: We apologize for this oversight. We have these contents in the revised manuscript.

Line 228: why old leaves? Mature leaves should be selected. It is known that leaves older than 5-6 weeks weaken their photosynthetic rate

Response: The details of old leaves have been supplemented in Figure S1. The descriptions in the manuscript have also been changed accordingly. Environmental stress caused great damage to the lower leaves of grapes. Therefore, we analyzed the upper and lower leaves separately to provide more accurate data for explaining the physiological changes of grape leaves under stress.

Reviewer 2 Report

Line 111: In subchapter 2.5 You should describe method for sugar analysis more detail.

Line 117.  With previous described methods of analysis (subchapter 2.6.) it is evident that You have an HPLC device. As a result, the question arises to why You did not use the HPLC for analysis, which is used as the gold standard in the analysis of anthocyanin content.

Line 122.  From the above description it is not clear how You estimated softening, especially how You graded softening according to the mentioned scale.

It is not shown in the paper, but the question is whether You measured the relative humidity, which significantly affects the grapevine maturation.

Author Response

Reviewer 2

Line 111: In subchapter 2.5 You should describe method for sugar analysis more detail.

Response: Sugar extraction and determination methods have been supplemented in section 2.6.

Line 117.  With previous described methods of analysis (subchapter 2.6.) it is evident that You have an HPLC device. As a result, the question arises to why You did not use the HPLC for analysis, which is used as the gold standard in the analysis of anthocyanin content.

Response: There are more than 20 kinds of grape anthocyanins. Thus, obtaining standard samples and qualitatively analyzing each chemical composition is difficult and almost impossible. Therefore, we did not use HPLC to detect the content of anthocyanins. The method used in this study can demonstrate differences in anthocyanin content between treatments and can serve as a good argument for the point of this manuscript.

Line 122.  From the above description it is not clear how You estimated softening, especially how You graded softening according to the mentioned scale.

Response: The fruit softening grading evaluation method is carried out with reference to the heat damage evaluation method (DOI: 10.5073/vitis.2018.57.75-81). The results of fruit softening evaluation and fruit firmness data corroborate each other, proving that our own method has certain feasibility.

It is not shown in the paper, but the question is whether You measured the relative humidity, which significantly affects the grapevine maturation.

Response: We observed no significant difference in relative humidity between treatments (Figure S2).

Reviewer 3 Report

The paper is very interesting and provides relevant data on the effect of different types of shade nets on vine physiology and grape composition.

The results concern only one year, i.e. one vegetative cycle. We know that the effect of microclimate modification has an influence on the reproductive cycle that takes place over two years. Thus, a minimum of two years should be considered to draw more robust conclusions.

The parameters concerning the composition of the grape are scarce (information about the acidity would be important).In addition, no data is presented on the effect of the nets on yield and yield components in the different treatments

Author Response

Reviewer 3

The paper is very interesting and provides relevant data on the effect of different types of shade nets on vine physiology and grape composition.

The results concern only one year, i.e. one vegetative cycle. We know that the effect of microclimate modification has an influence on the reproductive cycle that takes place over two years. Thus, a minimum of two years should be considered to draw more robust conclusions.

Response: Although the experiment in this manuscript has only 1 year of data, the experiment was carried out in the same year at two places (PQ and ZH). The research results obtained for both the places are common, and thus, the data are credible. In our follow-up study, we will also repeat the experiments in different years.

The parameters concerning the composition of the grape are scarce (information about the acidity would be important).In addition, no data is presented on the effect of the nets on yield and yield components in the different treatments

Response: There was no significant difference in the titratable acid content of the different treatments (Fig. R1); thus, no further analysis was performed. In addition, the premise of our experiment is to control the yield to 1000 kg/667 m2. Related information has been added in the revised manuscript.

(The figure cannot be displayed, please see the word file.)

Fig. R1 Titratable acid content across the different treatments at PQ and ZH. NONE: no shading; GRAY, BLUE, GREEN, and BLACK: shade net colors

Round 2

Reviewer 1 Report

Dear authors,

Thank you for clarifying some key points. Specifically, it is clear which leaves you have examined. However, I think you should clarify in the paper that the bottom leaves are usually impacted by stress.

In your reply, you made a very nice synthesis of your results, which I had found confusing. Why don't you report that synthesis in the Conclusions?

The Supplementary are very welcome.

Lines 82-83: it's not very clear; could you expand this concept?

Author Response

Dear reviewer,

Thank you for your comments and reminders.

The language of the article has been edited and revised by a native speaker. The information of standard viticulture and disease control practices and Conclusions section have been added and marked in red.

In your reply, you made a very nice synthesis of your results, which I had found confusing. Why don't you report that synthesis in the Conclusions?

Response: Conclusions have been added in the revised manuscript.

Lines 82-83: it's not very clear; could you expand this concept?

Response: The information of standard viticulture and disease control practices have been added in the revised manuscript.

Reviewer 2 Report

You have taken into account most of reviewers' suggestions, and I think that the paper is ready for publication.

Author Response

Dear reviewer,

Thank you for your comments and reminders.

The language of the article has been edited and revised by a native speaker.

Qian Zha